Prevalence and risk factors for falls among community-dwelling adults in Riyadh area

Alenazi Aqeel M. aqeelalenazi.pt@gmail.com 1
Alanazi Maram F. 2
Elnaggar Ragab K. 1 3
Alshehri Mohammed M. 4
Alqahtani Bader A. 1
Alhowimel Ahmed S. 1
Alhwoaimel Norah A. 1
Alanazi Ahmad D. 5
Alotaibi Mazyad A. 1
Almutairi Sattam M. 6
Alghamdi Mohammed S. 7
Bindawas Saad M. 8
1 Health and Rehabilitation Sciences, Prince Sattam Bin Abdulaziz University , Alkharj , Riyadh , Saudi Arabia
2 Occupational Therapy, Sydney University , Sydney , New South Wales , Australia
3 Department of Physical Therapy for Pediatrics, Cairo University , Cairo , Giza , Egypt
4 Physical Therapy, Jazan University , Jazan , Jazan , Saudi Arabia
5 Department of Rehabilitation Science, Majmaah University , Majmaah , Riyadh , Saudi Arabia
6 Department of Physical Therapy, College of Medical Rehabilitation, Qassim University , Buraydah , Qassim , Saudi Arabia
7 Department of Medical Rehabilitation Sciences, Umm Al-Qura University , Makkah , Saudi Arabia
8 Department of Rehabilitation Sciences, King Saud University , Riyadh , Saudi Arabia
Khosravi Mohsen
Electronic publication date: 2023 Dec 6
Publication date: 2023
Volume: 11
Electronic Location ID: e16478
Received 2023 Jul 19; Accepted 2023 Oct 26
Copyright: ©2023 Alenazi et al.
Copyright year: 2023
Copyright holder: Alenazi et al.
License: This is an open access article distributed under the terms of the Creative Commons Attribution License, which permits unrestricted use, distribution, reproduction and adaptation in any medium and for any purpose provided that it is properly attributed. For attribution, the original author(s), title, publication source (PeerJ) and either DOI or URL of the article must be cited.
License URL: https://creativecommons.org/licenses/by/4.0/

Keywords: Falling, Mental health, Falls, Middle aged, Elderly, Depressive symptoms, Chronic disease, Chronic illness

Funding: Prince Sattam Bin Abdulaziz University project number PSAU/2023/R/1445 This study was supported by funding from Prince Sattam Bin Abdulaziz University project number (PSAU/2023/R/1445). The funders had no role in study design, data collection and analysis, decision to publish, or preparation of the manuscript.

==============================
Objectives

This study aimed to assess fall prevalence, identify related risk factors, and establish cut-off scores for fall risk measures among community-dwelling adults in Riyadh region of Saudi Arabia.

Methods

A cross-sectional study was conducted in community, Riyadh city, Saudi Arabia. A sample of 276 Saudi citizens aged ≥40 years who were able to read and write in Arabic. Fall history and number of falls in the past 12 months were determined via self-reports. Variables assessed included demographic information, self-reported chronic diseases, depressive symptoms, and back pain severity.

Results

Participants were classified as either fallers (n = 28, 10.14%) or non-fallers. Fallers were more likely to have arthritis (odds ratio [OR]: 7.60, p = 0.001), back pain (OR: 5.22, p = 0.002), and higher depressive symptom scores (OR: 1.09, p = 0.013) than non-fallers. The number of reported falls was significantly associated with an elevated body mass index (incidence rate ratio [IRR]: 1.09, p = 0.045), arthritis (IRR: 8.74, p < 0.001), back pain (IRR: 4.08, p = 0.005), neurological diseases (IRR: 13.75, p < 0.007), and depressive symptoms (IRR: 1.08, p = 0.005). Cut-off scores predictive of falls associated with back pain and depressive symptoms were 1.5 (sensitivity: 0.61; specificity: 0.79; area under the curve [AUC]: 0.70) and 11.5 score (sensitivity: 0.57; specificity: 0.76; AUC: 0.66), respectively.

Conclusions

The prevalence of falls was relatively low among the individuals considered in this study. Chronic conditions, back pain severity, and depressive symptoms were determined to be associated with falls among community-dwelling individuals in Saudi Arabia.

Background

Falls are widely recognized for their detrimental effects on individuals’ health and well-being. The World Health Organization reported that falls are the second leading cause of injury-related deaths, contributing to an estimated 684,000 fall-related deaths worldwide (WHO, 2021). Although not all falls cause injuries, existing evidence suggests that one in five falls result in serious injuries, including head injuries and fractures (Alexander, Rivara & Wolf, 1992). The annual cost of medical care for fall-related injuries is substantial, with estimated medical costs for fall-related injuries as much as $50 billion in the United States (Florence et al., 2018). The Global Burden of Disease Study 2017 reported that the age-standardized prevalence of falls is 5,186 per 100,000 (James et al., 2020). Collectively, falls not only pose significant risks to an individual’s health and well-being but also exert a substantial burden on families, healthcare systems, and society at large.

In Saudi Arabia, research indicates that a significant proportion of older adults experience falls, with estimates ranging from 44.2% to 57.7% (Saif, Waly & Alsenany, 2012; Alshammari et al., 2018). A systematic review revealed that the pooled prevalence of falls among older adults residing in Gulf Cooperation Council (GCC) countries, including Saudi Arabia, is approximately 46.9% (Alqahtani et al., 2019). Additionally, it is noteworthy that the percentage of elderly individuals in Saudi Arabia has been steadily increasing. In the year 2000, the elderly population in the country was approximately 5.8%, and forecasts indicate that it is expected to reach 8.7% by 2026 and a significant 15% by 2050 (Nation U, 2020). As supported by current research, the likelihood of experiencing falls tends to rise as individuals age (Kannus et al., 2007). Considering this pattern and the projected expansion of the elderly population in Saudi Arabia, it becomes imperative to prioritize the implementation of preventive measures aimed at mitigating the adverse effects of falls within this demographic.

Existing evidence suggests that falling among middle-aged adults and older adults may depend on the complex relationship between both intrinsic and extrinsic factors for falls (WHO, 2021; Cuevas-Trisan, 2017; Formiga et al., 2008). There is emerging evidence identified risk factors for falls among middle-aged adults in the general population (Peeters et al., 2018; Peeters et al., 2019). Peeters and colleagues conducted co-ordinated analyses of cohort studies in four countries (Australia, Ireland, the Netherlands, and Britain) (Peeters et al., 2018; Peeters et al., 2019). When pooling data from all cohorts, poor mobility and urinary incontinence were identified as risk factors for falls for older individuals aged between 40 to 64 years old. Nevertheless, the authors noted that majority of investigated risk factors (demographic, health factors, and lifestyle) were associated with fall risk in at least one cohort.

In the elderly, several risk factors for falls have been reported including age, sex, poor balance related to medical conditions, medications, and unsafe environments (WHO, 2021; Cuevas-Trisan, 2017; Alenazi et al., 2018). However, the relative contributions of identified factors to overall fall risk remains controversial. Differences in reported values may be due to differences in study methodologies (such as sample size) and characteristics of the populations being studied. For example, inconsistent findings have been reported regarding whether demographic characteristics of the elderly are associated with falls. In several studies, sex has been reported as a risk factor, with women more likely to experience falls than men  (Gale, Cooper & Aihie Sayer, 2016). In contrast, Almegbel et al. (2018) reported that age and sex were not predictors of falls in a sample of individuals aged ≥ 60 years in Saudi Arabia. Fall risk increases when an individual is exposed to multiple risk factors (Dionyssiotis, 2012; Sharif et al., 2018). In a recent meta-analysis, Jehu et al. (2020) categorized risk factors into multiple domains to assess whether the domains were associated with recurrent falls. They found that medications and the presence of balance, mobility, psychological, sensory, and neuromuscular abnormalities are associated with recurrent falls in adults aged ≥ 60 years. Few studies examined the prevalence of fall including participants in the middle age and older adults (Talbot et al., 2005; Richardson, Bennett & Kenny, 2015). The prevalence of fall reported in these studies were 21% and 23.1%, respectively (Talbot et al., 2005; Richardson, Bennett & Kenny, 2015). Therefore, it is crucial to investigate risk of fall in this population.

While extensive research on falls and associated risk factors among elderly populations in various countries exists, there is a noticeable scarcity of evidence concerning the prevalence of falls and the associated risk factors among the general population of Saudi Arabia. While a few studies have reported the prevalence of falls and associated risk factors in Saudi Arabia (Alqahtani et al., 2019; Alosaimi et al., 2023; Alenazi, 2023), these studies predominantly focused on participants aged 50 or 60 years and older. The prevalence of chronic diseases has increased in both the elderly and the young population in Saudi Arabia (Alenazi et al., 2021; Alenazi & Alqahtani, 2023; Alqahtani et al., 2022; Alqahtani et al., 2021b; Alqahtani et al., 2020; Alqahtani et al., 2019). This is due to a number of factors, including urbanization and sedentary lifestyles  (Alqahtani et al., 2021a). It is essential to include middle-aged adults in the analysis to distinguish potential variations in prevalence and risk factors compared to older adults. Therefore, it is important to examine the prevalence and risk factors for falls in this population. This will ultimately help identify associated risk factors, establish health care priorities, and develop prevention and treatment strategies to for this population. Hence, the aims of this study were as follows: (1) to examine the prevalence of falls, (2) to investigate risk factors for falls (fallers versus non-fallers), (3) to explore the association between risk factors and number of falls, and (4) to determine cut-off scores for risk factors for falls (fallers versus non-fallers) among community-dwelling adults in Saudi Arabia. We hypothesized that specific demographic factors (i.e., older age, female sex, higher body mass index (BMI)) and the presence of chronic conditions would be associated with falls in this population.

Methods

Study design

Portions of this text were previously published as part of a preprint (Alenazi et al., 2023). This was a cross-sectional study using data obtained from community-dwelling adults in Riyadh and surrounding areas in Saudi Arabia. This study was approved by the research ethics committee (No. RHPT/019/031) from Prince Sattam Bin Abdulaziz University. All participants who agreed to participate in the study signed an informed written consent form prior to data collection.

Data collection procedures

Adult community-dwellers were recruited to participate in the study. Participants were eligible if they were aged ≥ 40 years, able to independently read and write in Arabic, and were Saudi citizens. Individuals were excluded if they were non-Saudi citizens or aged <40 years. Data were collected between October 2019 and April 2020 by research assistants trained in data collection.

Data were collected using a three-part, standardized form designed by the research team for the purpose of the study. The first part of the form contained questions related to demographics. The second part of the form aimed to gather data on the presence of chronic conditions using self-reported yes or no answers. Part three of the form included a question related to the fall history and number of falls. All participants filled out structured self-reported survey. Some of the items such as chronic diseases were verified by trained interviewer. All research assistants were trained on data collection. The validation process was taken place through randomly validating the results with some of the included participants through phone call. Participants were recruited from regions across Riyadh area including Riyadh city and Alkharj city from the community including mosques, malls, and other locations. These locations were selected due to the main focus on the community adults instead of patients visiting hospitals. Our previous observation indicated that the prevalence of falls from data from hospitals and clinics (27.77%) (Alenazi et al., 2023) is higher than that in the community (12.6%) (Alosaimi et al., 2023; Alenazi, 2023). All data were collected via written questionnaires or interviews.

Demographics

Demographic data were included including age, sex, weight, height, BMI, marital status, educational level, employment status, and smoking status. Age was recorded in years and sex was dichotomized into males and females. Weight was recorded in kilograms (KG) and height was recorded in centimeters (cm). Self-reported values from height and weight were used to calculate BMI by dividing weight in kg by height in m2. Marital status was recorded as single, married, divorced, and widowed. Educational level was categorized into none, elementary school, middle school, high school, and university. Employment status has three categories including unemployed, employed, and retired. Finally, smoking status was recorded as current smoker or non-smoker.

Chronic diseases

All data regarding chronic diseases were included using self-reported measures for the following conditions: arthritis, diabetes, hypertension, cardiovascular diseases, dyslipidemia, anemia, osteoporosis, neurological conditions, history of fracture, and back pain. If the participant has a disease not listed in the list, he/she can select others and name the disease. Self-reported measures for chronic conditions have good-to-excellent sensitivity and specificity  (Kehoe et al., 1994; Schneider et al., 2012; Bombard et al., 2005; Mannion et al., 2007).

Pain severity

For all participants, including those with back pain and arthritis, pain severity was obtained using a numeric pain rating scale that ranged from “0”, indicating no pain, to “10”, indicating severe pain.

Fall history

Fall history was reported by asking each participant the following two questions: (1) Have you experienced a fall in the past 12 months? (2) The number of falls (if a participant reported falling in the past 12 months). Participants categorized into fallers (no fall) and fallers (at least one fall). In this study, falls were defined according to the following World Health Organization description: “an event that results in a person coming to rest inadvertently on the ground, floor, or other lower-level” (WHO, 2021).

Depressive symptoms

Depressive symptoms were assessed using the Arabic version of the Beck Depression Inventory (BDI). The BDI is a widely used scale that includes 21 items, each rated from 0 to 3. Higher scores indicate the presence of depressive symptoms severe enough to require further clinical consideration. Psychometric properties were established for the Arabic version (Abdel-Khalek, 1998).

Sample size calculation

The sample has been determined based on previous research related to the prevalence of fall including participants in the middle age and older adults (Talbot et al., 2005; Richardson, Bennett & Kenny, 2015). The prevalence of fall reported in these studies were 21% and 23.1%, respectively. Therefore, we selected 23% prevalence for sample size calculation using this formula [N = Z2 P(1-P)/d2]. Where N = sample size, Z = Z score statistic for a level of confidence (1.96), P = the prevalence of falls in the middle age and older adults (23%), and d = the degree of precision (0.05). The sample size was estimated to be 272.

Statistical analyses

The primary outcomes of this study were fall history and number of falls. Comparisons between fallers and non-fallers were made using chi-square or Fisher’s exact tests for categorical variables and the independent t-test for continuous variables. A multiple binary logistic regression analysis was used to determine the association between risk factors and fall history (fallers versus non-fallers) using enter method to account for all variables together within the same model. The selection of entered variables was based on both clinical relevance to falls and significance level in the unadjusted model. Risk factors were entered to the model as predictors and fall status (yes for fallers versus no for non-fallers) was entered as the dependent variable (outcome). Odds ratios (OR) with 95% confidence intervals (95% CIs) were calculated for each risk factor. The primary analysis controlled for all confounders, including age, sex, BMI, marital status, education, employment status, smoking status, and the presence of chronic conditions (i.e., arthritis, diabetes, hypertension, cardiovascular diseases, dyslipidemia, anemia, osteoporosis, history of fractures, neurological disease, back pain, and depressive symptoms). In other words, the reported OR for risk factors was performed after controlling for all other reported variables. Missing variables were handled via case-wise deletion.

To identify associations between risk factors and number of falls, we used multiple negative binomial regression analyses. Incidence rate ratios (IRRs) and 95% CIs were calculated. This analysis used enter method to account for all variables together within the same model. Risk factors were entered to the model as predictors and number of falls (0, 1, 2. etc.) was entered as the dependent variable (outcome variable). The primary analysis was adjusted for all confounders including demographic details continuous (age and BMI) and categorical variables (sex, marital status, education, employment status, smoking, and chronic conditions [yes/no, i.e., arthritis, diabetes, hypertension, cardiovascular diseases, dyslipidemia, anemia, osteoporosis, history of fractures, neurological diseases, back pain]). Depressive symptom parameter BDI was considered as continuous variable.

To determine cut-off scores for significant risk factors of falls reported as continuous variables (i.e., back pain severity and BDI), a receiver operating characteristic (ROC) curve was used, and the area under the ROC was used to determine the overall accuracy of the model for predicting the occurrence of falls. The Youden index (sensitivity + [1 − specificity]) values were calculated to determine the best cut-off score with the largest Youden index. Sensitivity and specificity were calculated, indicating true positive and true negative results, respectively. An alpha level of 0.05 was used for all analyses. All analyses were performed using IBM SPSS for Mac, version 25.0 (SPSS Inc., Chicago, IL, USA).

Results

The final analysis included 276 participants. Participants with a history of at least one fall were classified as fallers (n = 28), and those who did not fall were classified as non-fallers (n = 248). The prevalence of falls among participants included in this study was 10.14%. The number of falls experienced in the past 12 months ranged from 1 to 9. Table 1 includes data describing demographic clinical characteristics of both fallers and non-fallers. Figure 1 shows the distribution of participants based on fall status. Of the potential risk factors assessed, only BMI, arthritis, hypertension, back pain, and depressive symptoms differed significantly when fallers and non-fallers were compared.

Table 1 Demographics and clinical characteristics of participants.

Factors	Fallers (n = 28)	Non fallers (n = 248)	p-value*	
Age, years (mean ±SD)	50.39 ± 11	49.11 ± 7	0.56	
Sex male/female	22/6	166/82	0.29	
BMI, KG/m2 (mean ±SD)	31.61 ± 5	28.48 ± 5	0.004	
Marital status			0.44	
Single	2	12		
Married	24	225		
Divorced	1	6		
Widowed	1	5		
Education			0.98	
None	1	15		
Elementary	2	16		
Middle	4	31		
High school	8	65		
University	13	121		
Employment status			0.63	
Unemployed	2	36		
Employed	20	162		
Retired	6	50		
Smoking	6	42	0.59	
Arthritis	11	27	<0.001	
Diabetes	5	44	0.58	
Hypertension	8	28	0.017	
Cardiovascular disease	2	7	0.23	
Back Pain	17	59	<0.001	
Dyslipidemia	3	28	0.61	
Anemia	2	13	0.65	
Osteoporosis	2	11	0.62	
History of fracture	10	25	0.001	
Neurological disease	2	6	0.19	
BDI Score (mean ± SD)	12.75 ± 9	7.59 ± 6	0.009	
Back pain severity (mean ± SD)	3.11 ± 3	1.00 ± 2	0.001	
Notes.

* p-value was based on Chi Square/Fisher’s Exact test for categorical variables or independent t-test for continuous variables.

SD Standard Deviation

BMI Body Mass Index

BDI Beck Depression Inventory

Figure 1 A flowchart for participants categories based on fall status.

Results of the binary logistic regression that examined the association between risk factors and falls (fallers versus non-fallers) are shown in Table 2, alongside calculated odds ratios (ORs) and associated 95% CI values. Fallers were more likely to have arthritis (OR: 7.60; 95% CI [2.27, 25.45]: p = 0.001), back pain (OR: 5.22; 95% CI [1.79, 15.22]; p = 0.002), and had more depressive symptoms (OR: 1.09; 95% CI [1.02, 1.16]; p = 0.013) than non-fallers.

Table 2 Binary logistic regression for history of falls (fallers and non-fallers) versus risk factors.

Factors*	OR (95% CI)	P-value	
Arthritis	7.60 (2.27, 25.45)	0.001	
Back Pain	5.22 (1.79, 15.22)	0.002	
Beck Depression Inventory	1.09 (1.02, 1.16)	0.013	
Notes.

OR odds ratio

Covariates included age, gender, education, employment status, marital status, smoking status, and all other chronic conditions.

* Only significant factors were reported.

Results of the negative binomial regression analysis examining the association between risk factors and the number of falls experienced by each patient, along with IRRs and associated 95% CIs are shown in Table 3. Number of falls was significantly associated with BMI (IRR: 1.09; 95% CI [1.01, 1.18]; p = 0.045), arthritis (IRR: 8.74; 95% CI [3.00, 25.85]; p < 0.001), back pain (IRR: 4.08; 95% CI [1.52, 10.98]; p = 0.005), neurological diseases (IRR: 13.75; 95% CI [2.06, 91.80]; p < 0.007), and depressive symptoms (IRR: 1.08; 95% CI [1.03, 1.14]; p = 0.005).

Table 3 Negative binomial regression for number of falls versus risk factors.

Factors*	IRR (95% CI)	P-value	
Body mass index	1.09 (1.01, 1.18)	0.045	
Arthritis	8.74 (3.00, 25.85)	<0.001	
Back Pain	4.08 (1.52, 10.98)	0.005	
Neurological disease	13.75 (2.06, 91.80)	0.007	
Beck Depression Inventory	1.08 (1.03, 1.14)	0.005	
Notes.

IRR Incidence rate ratio

Covariates included age, gender, education, employment status, marital status, smoking status, and all other chronic conditions.

* Only significant factors were reported.

Since back pain and depressive symptom severity were measured using the numeric rating scale and BDI, respectively, these data were comprised of continuous variables. Therefore, ROC curves were utilized to determine cut-off scores. ROC curve data used to determine cut-off scores for back pain and depressive symptoms are shown in Table 4. Using Youden’s index, we determined that the cut-off score for back pain was 1.5 (sensitivity, 0.61; specificity, 0.79), with an area under the curve (AUC) of 0.70, as shown in Fig. 2. The cut-off score for BDI was 11.5 (sensitivity, 0.57; specificity, 0.76), with an AUC of 0.66, as shown in Fig. 3.

Table 4 ROC curve for back pain severity scores and Beck Depression Inventory scores.

Variables	AUC (95% CI)	
Back pain severity (0–10)	0.70 (0.59, 0.82)	
Beck Depression Inventory	0.66 (0.55, 0.78)	

Figure 2 A receiver operating characteristic (ROC) curve for back pain severity associated with falls is shown.

Figure 3 A receiver operating characteristic (ROC) curve for the Beck Depression Inventory score associated with falls is shown.

Discussion

This study examined the prevalence of falls and their associated risk factors among community-dwelling adult citizens of Saudi Arabia. The prevalence of falls in the current study was 10.14%, which is much lower than that which was reported among middle aged populations in Germany (Schumacher et al., 2014) and Australia (Wang et al., 2021) (12.1% and 33%, respectively). Further, the prevalence of falls in the current study was also lower than the occurrence of fall in the older age population in Turkey (Cevizci et al., 2015), Hong Kong (Chu, Chi & Chiu, 2005), the UK (Gale, Cooper & Aihie Sayer, 2016), and the US (Jia et al., 2019) (32%, 19.3%, 28%, and 23%, respectively). Interestingly, local studies also reported that the prevalence of falls is higher than that which was reported here(46.9%, 49.9%, and 57.7%) (Alshammari et al., 2018; Alqahtani et al., 2019; Almegbel et al., 2018). A possible explanation for this discrepancy is that our sample included a large range of age groups (individuals aged 40–80 years), whereas other local studies included older adults (aged > 60 years). The average age of participants in the current study was approximately 50 years, which is lower than the average age reported in prior studies. Moreover, the large variation between sample sizes and differences in the geographical areas within which studies have taken place may contribute to observed differences in fall prevalence values reported in Saudi Arabia.

Our study showed that fallers were eight times more likely to have arthritis than non-fallers, indicating that arthritis highly impacts fall risk. Our findings are in line with those of several other studies, which reported that arthritis increases the risk of falling among middle-aged population (Ofori-Asenso, Ackerman & Soh, 2021). Two studies that used a self-reported diagnosis method for arthritis that did not require participants to specify the arthritis site, similar to that which was used in the current study, reported that arthritis increased risk of multiple falls (Arden et al., 1999) and was associated with number of falls (Prieto-Alhambra et al., 2013). Other studies that used clinical and radiographic definitions of arthritis reported a similar association between arthritis and falls (Muraki et al., 2011; Van Schoor et al., 2020). Several factors may explain the association between arthritis and falls. Pain and pain medications have been reported to be potential mediators of this relationship (Lo-Ciganic et al., 2017). Furthermore, low physical performance metrics including impaired dynamic balance, gait speed, muscle weakness, and decreased mobility, have been identified as potential mediators of the association between arthritis and falls  (Stel et al., 2003; Cöster et al., 2020; Peeters et al., 2010).

Our data indicating that back pain is associated with fall risk are consistent with that of previous research (Muraki et al., 2011; Marshall et al., 2016). Musculoskeletal pain, including back pain, has been reported to increase fall risk  (Marshall et al., 2016; Marshall et al., 2017; Patel et al., 2013; Kimachi et al., 2019; Leveille et al., 2009; Woo, Leung & Lau, 2009; Muraki et al., 2013). Further, its prevalence increases with age (Kimachi et al., 2019; Hartvigsen et al., 2018; Castro et al., 2020). The cut-off score for predicting fall risk was modest (1.5 using the numeric pain scale), which may be explained by the small sample size and low degree of back pain score variability reported among fallers in the study. Although several studies have reported an association between back pain and falls, few have explored the nature of this association. For example, it has been suggested that pain alters gait patterns and speed, which may explain increases in fall risk among those with back pain (Simonsick et al., 2018). Structural changes have been proposed by Ito et al. (2019) who reported that a decreased cross-sectional lumbar multifidus muscle area increases the risk of falls. A clinical implication of that work is that addressing back muscle strength may serve as an effective preventive measure against falls.

Depressive symptoms are among the best defined risk factors for falls  (Ouyang & Sun, 2018) . Unexplained falls with no prior incident are more likely to occur in people with depressive symptoms than in those without (Briggs, Kennelly & Kenny, 2018). Our study suggests that the presence of depressive symptoms, as defined by the BDI, is a risk factor for falls among the individuals considered. However, this finding should be interpreted with caution given that the OR value for depressive symptoms was small (1.09) compared to those for arthritis (7.6) and back pain (5.22)-associated variables. The cut-off score for using the presence of a current episode of major depressive symptoms for differentiating fallers from non-fallers was determined to be 11.5 (Suija et al., 2012). In addition, Rose et al. (2015) proposed a cut-off value for severe depressive symptoms of between 20 and 24 for men, and 23 and 28 for women. However, the cut-off score (i.e., 11.5 using BDI) needs to be assessed using controlled studies among a variety of populations, to better understand whether other factors contribute to BDI. After controlling for common mechanisms related to falls in this study, depressive symptoms were determined to be associated with fall risk. However, factors that mediate the association between fall risk and depressive symptoms require further investigation, if knowledge of these underlying mechanisms is to be used to optimize preventive strategies and treatment plans for patients at risk of falls.

Our study showed that when accounting for demographic variables, the presence of neurological diseases, arthritis, and back pain, were associated with an increased fall frequency. Though BMI and depressive symptoms were significantly associated with fall number, these data are not necessarily clinically important given that the IRR values were close to 1. The finding that neurological diseases are associated with a high number of falls was expected. However, its high IRR value of 13.75 and wide confidence interval might not accurately reflect this association. This was primarily attributed to the fact that relatively few participants reported neurological diseases and the number of falls they experienced.

This study had some limitations that should be addressed in future studies. Investigating risk factors of risk of falls using a larger sample size may help minimize the risk of type 2 errors. Although this study categorized fallers two ways (fallers vs. non-fallers and number of falls), adding another category that considers injuriousness as an outcome measure might increase our understanding of fall risk. The design of this study was cross-sectional, which limits our ability to determine causality. For instance, we still do not know if depressive symptoms increased fall risk, or if recurrent falls increased the risk of depressive symptoms. This study used a convenient small sample from the community. Due to limited resources and time, the data was collected around the Riyadh region. Many challenges appeared during participants recruitment including the beginning of COVID-19 and lack of interest from participants to enroll. As there is no optimum method for measuring fall risk, there is a possibility of recall bias regarding the number of falls reported. Therefore, the validity of the findings needs to be considered in future work. The sample size in the fallers group is small, and this might be attributed to the inclusion of middle-aged individuals. This study used self-reported measurements of comorbidities, depressive symptoms, and pain symptoms. Several risk factors for falls including antidepressant medication use, functional capacity, and cognition, were not considered in this study. We used IRR for data analysis for number of falls in the current study while this approach fits best a prospective design.

Conclusion and Future Research

The prevalence of falls was reported to be 10.14% in this study, a value less than that reported for the general Saudi population. Several risk factors were associated with fall occurrence, including obesity, arthritis, hypertension, back pain, and depressive symptoms. Increased number of falls was associated with an elevated BMI, presence of arthritis, back pain, neurological disease, and increased depressive symptom severity using BDI in this population. Future research using structured diagnostic tools is needed to improve study validity and reliability. Further, there is a need to establish a longitudinal study that may be used to improve our understanding of the direction of the identified associations to formulate better fall prevention and treatment strategies. A future investigation of multidimensional determinants of falls in older adults that uses biopsychosocial models to guide our understanding of the problem is warranted.

Supplemental Information

Supplemental Information 1 Prevalence and risk factors for falls among community-dwelling adults

Dataset ID doesn’t contain any identifiable initials for the participants.

Click here for additional data file.

Supplemental Information 2 Prevalence and risk factors for falls among community-dwelling adults

Dataset ID doesn’t contain any identifiable initials for the participants.

Click here for additional data file.

Supplemental Information 3 Codebook for categorical variables

Click here for additional data file.

All authors acknowledge the students and participants.

Abbreviation

GCC Gulf Cooperation Council

BMI Body Mass Index

BDI Beck Depression Inventory

OR odds ratio

IRR Incidence Rate Ratio

AUC Area Under the Curve

Additional Information and Declarations

Competing Interests

Author Contributions

Human Ethics

Data Availability

The authors declare there are no competing interests.

Aqeel M. Alenazi conceived and designed the experiments, performed the experiments, analyzed the data, prepared figures and/or tables, authored or reviewed drafts of the article, and approved the final draft.

Maram F. Alanazi performed the experiments, prepared figures and/or tables, authored or reviewed drafts of the article, and approved the final draft.

Ragab K. Elnaggar performed the experiments, authored or reviewed drafts of the article, and approved the final draft.

Mohammed M. Alshehri conceived and designed the experiments, performed the experiments, authored or reviewed drafts of the article, and approved the final draft.

Bader A. Alqahtani conceived and designed the experiments, performed the experiments, authored or reviewed drafts of the article, and approved the final draft.

Ahmed S. Alhowimel conceived and designed the experiments, performed the experiments, authored or reviewed drafts of the article, and approved the final draft.

Norah A. Alhwoaimel conceived and designed the experiments, performed the experiments, authored or reviewed drafts of the article, and approved the final draft.

Ahmad D. Alanazi performed the experiments, authored or reviewed drafts of the article, and approved the final draft.

Mazyad A. Alotaibi performed the experiments, authored or reviewed drafts of the article, and approved the final draft.

Sattam M. Almutairi performed the experiments, analyzed the data, authored or reviewed drafts of the article, and approved the final draft.

Mohammed S. Alghamdi performed the experiments, analyzed the data, authored or reviewed drafts of the article, and approved the final draft.

Saad M. Bindawas conceived and designed the experiments, performed the experiments, analyzed the data, authored or reviewed drafts of the article, and approved the final draft.

The following information was supplied relating to ethical approvals (i.e., approving body and any reference numbers):

This study was approved by the research ethics committee (No. RHPT/019/031) from Prince Sattam Bin Abdulaziz University.

The following information was supplied regarding data availability:

The raw data are available in the Supplemental Files.

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
