# Peer review of "Prevalence and risk factors for falls among community-dwelling adults in Riyadh area"

_PeerJ, doi:10.7717/peerj.16478_

## Round 0.1 · original submission · Major Revisions

I have now received the reviewers' comments on your manuscript. They have suggested some revisions to your manuscript. Therefore, I invite you to respond to the reviewers' comments and revise your manuscript.

**Language Note:** PeerJ staff have identified that the English language needs to be improved. When you prepare your next revision, please either (i) have a colleague who is proficient in English and familiar with the subject matter review your manuscript, or (ii) contact a professional editing service to review your manuscript. PeerJ can provide language editing services - you can contact us at copyediting@peerj.com for pricing (be sure to provide your manuscript number and title). – PeerJ Staff

Reviewer 1 ·

Basic reporting

By dividing the Background section into Introduction and Related Research, Introduction should focus on research background and research motivation, and Related Research should focus on previous studies related to this study.

Experimental design

It is somewhat unreasonable to generalize research results in small populations and limited areas..Please describe in detail why this study collected data only in a small population and Riyadh region.

Also, describe whether the questionnaire was self-reported by the survey participants, and whether they responded truthfully.

In addition, arthritis, back pain, and depression are thought to be common and predictable causes of falls. Explain why you didn't consider more diverse diseases.

Validity of the findings

no comment

Additional comments

The last paragraph of the Discussion section (line 312-328) should be described as a future research works. Therefore, it is recommended that the Conclusion section will be changed to 'Conclusion and Further Research Works' and that future research tasks will be described here.

In addition, since the survey was limited to Riyadh region-centered, it is recommended to change the title of the paper from 'Saudi Arabia' to 'Riyadh Area'.

·

Basic reporting

Thank you so much for letting me review this article
Title
May I suggest changing it?
Prevalence and risk factors for falls among community-dwelling adults in Saudi Arabia
And add age to the abstract
Abstract section
The background heading needs to be changed to reflect the aim of the objective.
Avoid abbreviations in the abstract
Keywords need to follow MESH guidelines
Introduction
Your target population was the adult age group that starts at 40; most of your introduction background signifies the study variables among the elderly. The authors need to focus on the study age groups.
I need help understanding what an hour of study adds to this area of research. You need to magnify the study variables, especially considering the age group.

Experimental design

Informed written consent must be added.
Where is the section on sample size calculation? I think it was missing
The authors need to add an equation for sample size calculation and use STROBE guidelines in writing the observation study.
A flow chart must be added to elaborate on the number of adults and categories of fallers and non-fallers.
Data collection
What are the places that authors collect data from?
Research assistant trained in data collection
How can authors validate the results of their data with their assistants?
Tools and measures need to be deleted, starting with the description of tools in the demographic.
What are the limitations of the study?

Validity of the findings

The sample size for fallers adults is minimal and can't represent a study in a country like Saudi Arabia.
Why did the authors not collect data from places where this type of person is commonly found, like hospitals and rehabilitation centers for post-fracture patients? That will increase the probability of increasing the sample size.
Also, the comparison between fallers and non-fallers could be more illogical, as the discrepancy between the numbers of both groups was too great.
The statistics used supported Your discussion and results, but it was still an issue. The problem of sample size needs to be more representative. The authors consider this study an observational prevalence study first made in Saudi Arabia.

Additional comments

no comment

---

## Round 0.2 · accepted · Accept

In my opinion this manuscript has been revised with attention to the reviewers' comments and can now be published.

Reviewer 1 ·

Basic reporting

The paper revision requirements have been reflected in the revised version.

Experimental design

The paper revision requirements have been reflected in the revised version.

Validity of the findings

The paper revision requirements have been reflected in the revised version.

Additional comments

No more comments